# Magnetron Deposition of Cr Coatings with RF-ICP Assistance

Dmitrii V. Sidelev *, Vladislav A. Grudinin, Konstantin A. Zinkovskii, Kamila Alkenova and Galina A. Bleykher

School of Nuclear Science and Engineering, Tomsk Polytechnic University, 30 Lenina av., 634050 Tomsk, Russia
* Correspondence: sidelevdv@tpu.ru; Tel.: +7-3822-70-17-77 (ext. 2518)

**Abstract:** The article describes a comparative analysis of chromium coatings deposited by magnetron sputtering with and without ion assistance induced by a radiofrequency inductively coupled plasma (RF-ICP) source. Four series of 2 μm-thick Cr coatings were prepared, and then their cross-sectional microstructure, crystal structure and corrosion resistance were investigated by scanning and transmission electron microscopy, X-ray diffraction and a potentiodynamic polarization method. RF-ICP assistance led to significant enhancement (almost twofold) of ion current density in a substrate. The role of RF-ICP assistance in coating properties for planetary-rotated substrates was defined in terms of ion-to-atom ratio in particle flux entering a substrate. Calculations of particle and ion flux densities revealed an increase in ion-to-atom ratio from 0.18 to 1.43 and 0.11 to 0.84 in substrate positions distant from the magnetron sputtering systems depending on their design. RF-ICP assistance is beneficial for depositing dense Cr coatings with increased corrosion resistance in a 3.5 wt.% NaCl solution. The corrosion rate of AISI 321 steel can be decreased from $6.2 \times 10^{-6}$ to $4.0 \times 10^{-8}$ mm/year by deposition of the dense Cr coating.

**Keywords:** magnetron sputtering; Cr coatings; corrosion resistance; RF-ICP assistance; cross-sectional microstructure; crystal structure





## 1. Introduction

Chromium coatings are often used to protect materials against corrosion [1–3], oxidation [4,5] and hydrogen embrittlement [6,7] in different aggressive environments. Electroplating is a well-known technology used to produce Cr coatings, which has high productivity and low cost of deposition process, conformal deposition onto complex-shaped substrates and ability to adjust properties of obtained coatings. Nevertheless, ion-plasma technologies are becoming more relevant to surface modification due to the extremely high toxicity of compounds involved in chromium electroplating [8] as well as difficult surface activation of some materials, which can easily form surface oxides (e.g., titanium, zirconium and their alloys) [9–11]. Magnetron sputtering is already applied for Cr coating deposition, providing relevant functional properties, high purity and adjustment of structural parameters of coatings. To date, many types of magnetron sputtering system (MS) have been used for industrial and scientific tasks [12]. However, despite many problems of magnetron sputtering having been solved, some critical drawbacks still exist. The first one is relatively low productivity, which is limited by target power density prior to target overheating. One possible solution consists of using MS with a target with thermal isolation from a water-cooled magnetron body. In this case, target sublimation or even evaporation can occur, which leads to an increase in total particle flux from target to substrate [4,13]. The second feature is the porous and fiber/columnar microstructure of magnetron-deposited coatings. Indeed, according to structure-zone models (SZM) [14,15], magnetron-deposited coatings usually have a columnar microstructure under typical deposition conditions. Some approaches can be used to enhance coating density and modify cross-sectional microstructure. These can be achieved by applying a high-density plasma obtained from high-power pulsed magnetron sputtering (HiPIMS) [16,17] or by modifying a magnetic system of MS to create an unbalanced configuration, enabling a high ion current

density of a substrate [18,19]. An additional approach can be also suggested to modify a coating microstructure such as ion assistance induced by radiofrequency inductively coupled plasma (RF-ICP). This leads to an increase in ion current and ionization degree of sputtered particle flux in a substrate [20,21]. However, RF-ICP sources usually have a spiral-shaped electrode, which is placed in a vacuum chamber. Due to this, sputtering of an electrode material leads to decreased coating purity and can influence coating properties. Nevertheless, some RF-ICP sources are designed with a planar spiral, which is taken from a vacuum chamber. In this case, a quartz window is usually used to transport RF radiation in a vacuum chamber [22,23], which can be beneficial for coating deposition. Intensive ion bombardment induced by RF-ICP assistance can be used to modify structural and functional properties of deposited coatings. Due to this, RF-ICP assistance should be used to deposit a metallic Cr coating in an Ar atmosphere, as this coating type is considered a candidate material for protection of Zr alloys in aggressive environments [4–7,10,11]. Thus, this preliminary study aims to determine the role of RF-ICP assistance in structural properties and corrosion resistance of Cr coatings obtained by magnetron sputtering. For this purpose, several Cr coating series were prepared using magnetron sputtering with and without RF-ICP assistance. Then, their crystal structure, cross-sectional microstructure and corrosion resistance were studied and then analyzed based on particle and ion flux densities in substrates.

## 2. Materials and Methods

### 2.1. Sample Preparation

Two designs of magnetron sputtering system (MS) were used for Cr coating deposition. The first one was a conventional system, where a Cr target was directly cooled by water. The second type of MS was equipped with a hot target (limited cooling). Cr targets with direct and limited cooling are further titled "conventional" and "hot", respectively. A more detailed description of considered designs of MS is presented in previous papers [6,13]. A schematic drawing of the experimental setup is shown in Figure 1. The MS was placed on the wall of the vacuum chamber, samples were fixed in a planetary-rotated substrate-holder with a diameter of 300 mm. The RF-ICP source (RPG-128 [22], Laboratory of Plasma Technologies Plus, LLC, Moscow, Russia) was installed on the top wall of the vacuum chamber and connected to the power source COMDEL CX1250 with an operation frequency of 13.56 MHz. The RPG-128 had a special protective shield preventing coating deposition on an output quartz window of the source.

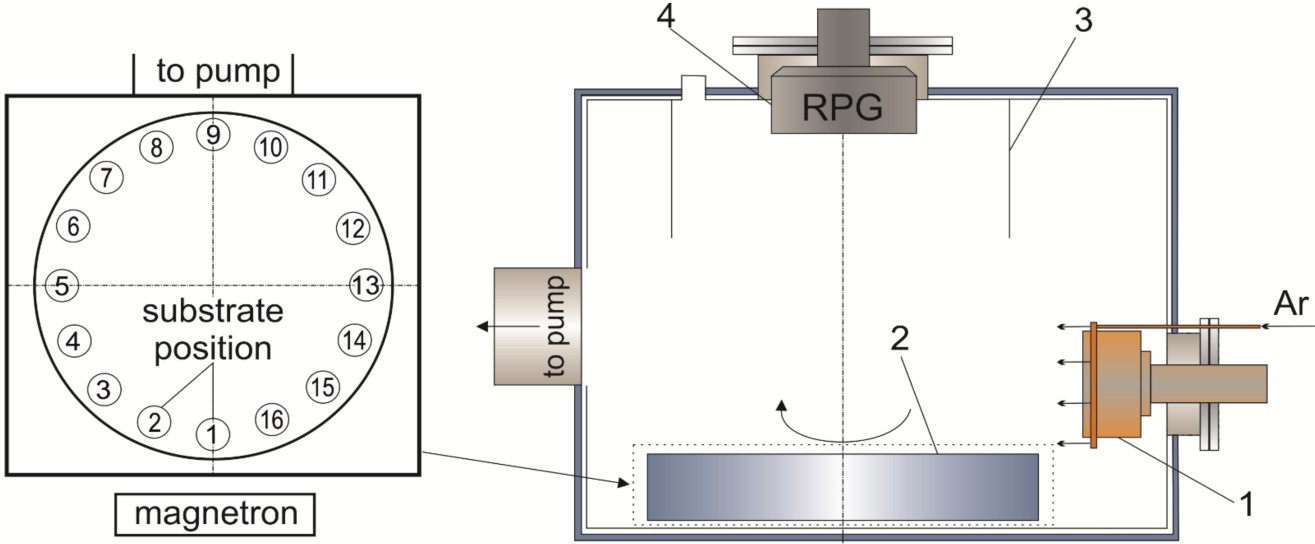

**Figure 1.** Schematic drawing of the experimental installation: 1—MS; 2—planetary-rotated substrate-holder; 3—protective shield; 4—RPG-128.

Cr coatings were deposited on Si (110) wafers and AISI 321 (C—0.8 wt.%, Mn—2.0 wt.%, P—0.045 wt.%, S—0.03 wt.%, Cr—18.0 wt.%, Ni—10.3 wt.%, Ti—0.52 wt.%, Si—0.75 wt.% and Fe balance) steel substrates. Steel substrates were grounded and polished to obtain an averaged surface roughness ($R_a$) of ~0.04 µm. Then, substrates were treated in an ultrasonic bath with isopropyl alcohol for 20 min.

The residual pressure in the vacuum chamber was equal to $5 \times 10^{-3}$ Pa in each deposition process. Before coating deposition, the surface of substrates was bombarded by $Ar^+$ using an ion source with closed electron drift (2.5 kV and 40 mA) for 10 min. Four coating series were prepared by magnetron sputtering of conventional and hot Cr targets with and without RF-ICP assistance. Deposition parameters of Cr coatings are presented in Table 1. The target power density and substrate bias potential were equal to 15.7 W/cm$^2$ and −50 V in all deposition modes, respectively. The power of the RF-ICP source was equal to 500 W. An infrared pyrometer (Optris CTlaser 3MH1CF4) operated at a spectral length of 2.3 µm was used to determine the temperature of samples during coating deposition.

**Table 1.** Deposition parameters of Cr coatings.

| Deposition Mode | Target Type | RPG-128 | $t$, min | $p$, Pa | $I_{bias}$, mA/cm$^2$ | $h$, µm | $T$, K |
|---|---|---|---|---|---|---|---|
| 1 | conventional | no use | 108 | | 1.1 | | 375 |
| 2 | | applied | 120 | | 1.9 | | 422 |
| 3 | hot | no use | 70 | 0.3 | 2.3 | 2.0 | 568 |
| 4 | | applied | 80 | | 3.7 | | 635 |

Note: $t$—deposition time; $p$—operation pressure; $I_{bias}$—ion current density to a substrate; $h$—coating thickness; $T$—maximal substrate temperature.

### 2.2. Calculation of Particle Flux and Ion Current Density to A Substrate

As Cr coating deposition was performed using the planetary rotation substrate-holder, particle flux and ion current density to a substrate were calculated depending on the substrate position. The mathematical model of the erosion process for magnetron sputtering of a solid-state target, taking into account sublimation and local evaporation, was used in this study [24]. Calculations of particle flux density ($F_{dep}$) were performed according to the second law of Lambert–Knudsen [25]. If target and substrate are flat and placed parallel to each other, and the origin of coordinates is in the plane coinciding with target surface, $F_{dep}$ can be calculated at any point on a substrate surface with coordinates $(x, y, L)$ using the following equation:

$$F_{dep}(x, y, L) = \frac{L^2}{\pi} \iint\limits_{S_t} \frac{F_{sum}(x_t, y_t) dx_t dy_t}{\left( L^2 + (x - x_t)^2 + (y - y_t)^2 \right)^2}, \tag{1}$$

where $S_t$—target surface area; $x_t$, $y_t$—coordinates of the center of any elementary area ($dx_t \cdot dy_t$) on target surface with a particle flux density $F_{sum}(x_t, y_t)$; $F_{sum}(x_t, y_t)$—total particle flux density on a substrate by target sputtering ($F_{sput}$) and sublimation ($F_{subl}$); $L$—distance between target and substrate.

The flux density of sputtered particles ($F_{sput}$) is proportional to the current density of ions ($I_{t,ion}$) extracted from MS plasma and to the sputtering yield ($S$):

$$F_{sput} = \frac{S \cdot I_{t,ion}}{q}, \tag{2}$$

where $q$—elemental charge. Sputtering yield depends on the energy of ions, which are accelerated in the electric field created in a discharge gap. This value is usually taken as $0.75 \cdot U$ [26], where $U$—discharge voltage in electrical circuit of MS. The distribution of ion current density ($I_{t,ion}$) along a radius of target surface ($r$) repeats the distribution of the horizontal component of the magnetic field near the target surface [24]. Sputtering yield is calculated in the SRIM code [27].

The flux density of sublimated atoms ($F_{subl}$) was calculated using the Hertz–Knudsen equation:

$$F_{subl} = \frac{1}{(2\pi \cdot m \cdot k \cdot T_t)^{1/2}} \cdot p_{sat}(T_t), \tag{3}$$

where *m*—atom mass; *k*—Boltzmann constant; $p_{sat}(T_t)$—saturated vapor pressure at surface temperature ($T_t$).

Prior to coating deposition, the planetary substrate-holder was fixed in this position as substrates were placed in the order shown in Figure 1. Substrates were separately connected to nine feedthrough terminals. Ion current in the substrate was determined in each position from #1 to #9 using a digital oscilloscope, Tektronix TDS 2022B, and the distribution of ion current density ($I_{bias}$) of substrates was calculated.

*2.3. Sample Characterization*

Scanning electron microscopy (SEM, Vega 3, Tescan, Brno, Czech Republic) in the operation mode of secondary electrons (SE) was used to determine the thickness and analyze a cross-sectional microstructure of obtained coatings. Detailed analysis was also carried out by transmission electron microscopy (TEM) on a JEF-2100 (JEOL, Tokyo, Japan) microscope. Crystal structure of Cr coatings was investigated by X-ray diffraction (XRD, Shimadzu XRD-7000S, Shimadzu, Kyoto, Japan) in a *2·θ* range from 10 to 90°. An X-ray tube with Cu-Kα radiation (40 kV, 30 mA) was used. Then, XRD diffractograms were analyzed using the PDF-4+ database and structural parameters (lattice parameter, crystallite size and texture coefficients) were calculated.

Corrosion resistance of Cr coatings was determined by a potentiodynamic polarization method in a 3.5 wt.% NaCl solution using a potentiostat–galvanostat P-45X equipped with a three-electrode electrochemical cell. Potentiodynamic polarization tests were performed at room temperature (~25 °C), and the working area of samples was equal to 0.5 cm². A reference electrode was AgCl electrode (4.2 M KCl), while a counter electrode was made from high-purity graphite. Before taking measurements, each sample was kept in the solution at open circuit potential for at least 2000 s to compensate charges. Then, potentiodynamic polarization tests were carried out in a potential range from −700 to 2000 mV with a scan rate of 0.5 mV/s.

## 3. Results

*3.1. Process Parameters*

The distribution of ion current density ($j_{bias}$) of substrates is plotted in Figure 2.

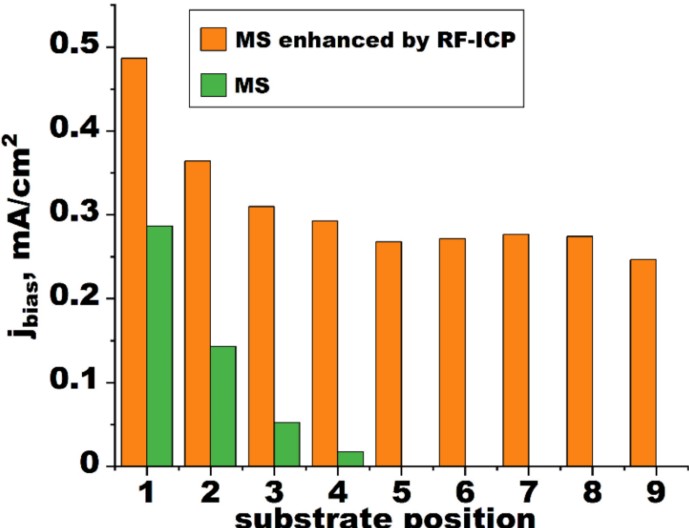

**Figure 2.** Ion current density ($I_{bias}$) of substrates depending on their position in the substrate-holder.

The distribution of ion current density depends on deposition modes. When the RF-ICP source was not used, ion current was detected only in the substrate position near the MS. The highest ion current density was in position #1 and then it noticeably decreased (pos. #4). No ion current was detected in positions that were distant from the MS (pos. #5 → pos. #9). However, when the RF-ICP source was applied, ion current was observed in all substrate positions. The highest ion current density was also found in the substrate position directly opposite to the MS (pos. #1). Then, $j_{bias}$ decreased to 0.28 mA/cm$^2$ (pos. #5) and did not change up to position #9, which indicates substrate bombardment by ions predominantly from a RF-ICP discharge. The obtained data also demonstrate that applying RF-ICP assistance can lead to a significant increase in ion current density of substrates. When the RF-ICP source was used, $j_{bias}$ increased in position #1 from 0.29 to 0.49 mA/cm$^2$.

The dependence of total particle flux density ($F_{dep}$) on substrate position is shown in Figure 3. Inputs of sputtering ($F_{sput}$) and sublimation ($F_{subl}$) in total particle flux density are also presented.

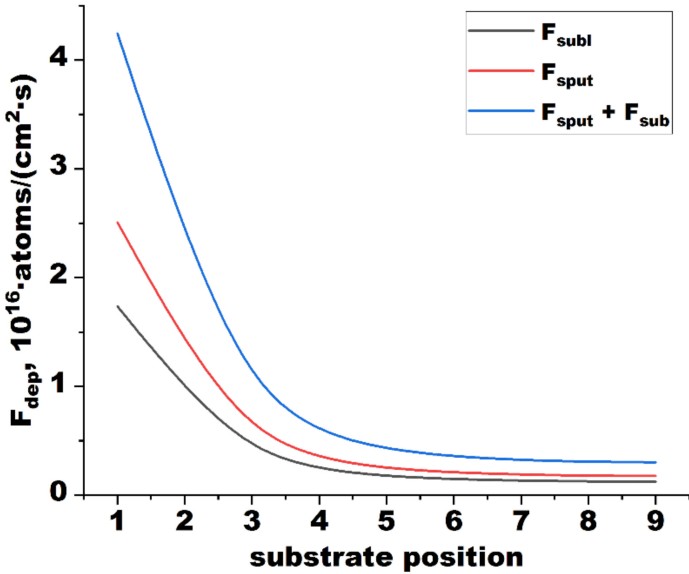

**Figure 3.** Total particle flux density ($F_{dep}$) to a substrate due to target sputtering ($F_{sput}$) and sublimation ($F_{subl}$) depending on substrate position.

Particle flux of a substrate comprises only sputtering particles in the case of using MS with a conventional Cr target. Thereby, total particle flux density of the substrate ($F_{dep}$) was equal to particle flux density by target sputtering ($F_{sput}$), which decreased for higher positioned numbers as the distance between target and substrate increased. The maximum value of $F_{dep}$ was observed in the substrate position (pos. #1) opposite to the MS (2.5 × 10$^{16}$ atoms/(cm$^2$·s)); then, it sharply decreased up to 3.4 × 10$^{15}$ atoms/(cm$^2$·s) in position #4. At a higher distance of substrates from the MS (pos. #4 → #9), $F_{dep}$ changed to a lesser extent.

Magnetron sputtering with hot Cr target included both process, target sputtering and sublimation. According to calculations, $F_{sput}$ was somewhat higher than that of the particle flux density of the substrate due to target sublimation. This ratio of $F_{sput}$ to $F_{subl}$ was determined by the applied target power density (15.7 W/cm$^2$) [24]. Thereby, $F_{dep}$ was higher for hot Cr target magnetron sputtering compared to conventional MS, which has also been observed in deposition conditions (Table 1). The dependence of $F_{dep}$ on substrate position had the same behavior as for conventional Cr target sputtering. It was equal to 4.2 × 10$^{16}$ atoms/(cm$^2$·s) in position #1 and then decreased up to 3.0 × 10$^{15}$ atoms/(cm$^2$·s) in position #9.

### 3.2. Structural Parameters

Figure 4 shows cross-sectional SEM microstructure images of Cr coatings deposited on Si (110) substrates.

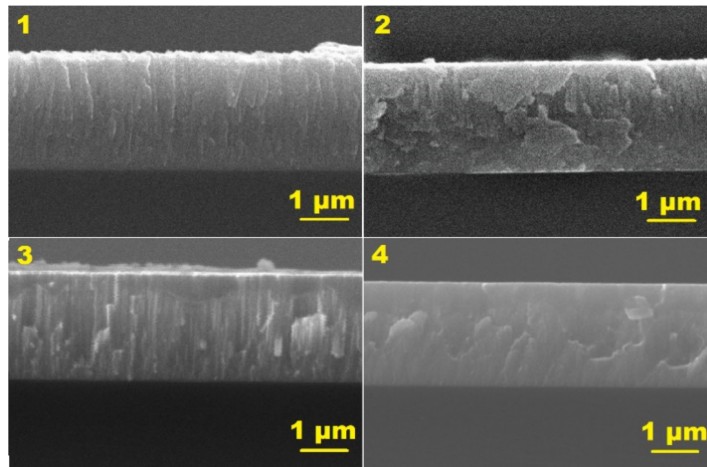

**Figure 4.** SEM images of cross-sectional microstructure of Cr coatings obtained on Si (110) substrates depending on their deposition modes: 1—dep. mode #1; 2—dep. mode #2; 3—dep. mode #3; 4—dep. mode #4.

All coatings have the same thickness (~2 μm) but a different cross-sectional microstructure depending on their deposition modes. Without RF-ICP assistance, a columnar microstructure of Cr coatings is well observed for both designs of the MS. According to SZM models [14,15], deposition conditions of these Cr coatings corresponded to zone I ($T/T_m = 0.17$ and 0.26, where $T_m$—melting temperature of a coating material, $T$—substrate temperature). This zone is characterized by low adatom mobility and forms a columnar microstructure. Despite a higher value of $T/T_m$ (0.26), the Cr coating obtained in the deposition mode #3 had a more pronounced columnar microstructure, which is due to the increased deposition rate caused by sublimation of hot Cr target. Applying RF-ICP assistance led to a densification of coating microstructure for both designs of the MS. Taking into account the not so high difference in $T/T_m$ for samples obtained with and without ion assistance as well as a low bias potential (−50 V) applied to substrates, observed changes in coating microstructure are primarily caused by enhanced ion bombardment by low-energy ions formed by the RF-ICP source. It resulted in increasing mobility of adatoms on substrate surface and "knock-on" effect [28]. To confirm this, Cr coating deposition on Zr alloy substrate was additionally performed using the MS with hot Cr target enhanced by the RF-ICP source (deposition mode #4).

TEM images of the cross-sectional microstructure reveal a dense microstructure of the deposited Cr coating without any voids in interface or coating regions (Figure 5). High-resolution TEM image (Figure 5b) shows well-defined lattices in the Cr coating region, while a light contrast indicates the uniform interface between Cr coating and Zr alloy substrate. According to the selected area diffraction (SAED) shown in Figure 5d, the obtained Cr coating had (110), (200) and (211) planes corresponding to a body-centered cubic (bcc) Cr phase (PDF #06-0694). The TEM image in the Cr coating region (Figure 5c) demonstrates a layered structure that is highlighted by yellow arrows. The thickness of one layer is equal to ~34 nm, and this layer thickness is observed through the whole cross-section of the Cr coating.

The crystal structure of Cr coatings was studied by XRD (Figure 6), and the structural parameters of coatings are presented in Table 2. XRD difractograms reveal a bcc structure of Cr coatings with (110), (200) and (211) orientations. Grain growth direction depended on deposition modes (Table 2). It was found that Cr coatings had the highest texture coefficient of (110) plane ($T_{C(110)}$~2.6), when samples were obtained using only the MS. As the RF-ICP

source was applied, a change in growth direction was found for predominant Cr (110) to Cr (211) orientation. Texture coeffients of Cr (211) became 2.8 and 1.7, while $T_{C(110)}$ decreased to 0.2 and 0.4 for deposition modes #2 and #4, respectively.

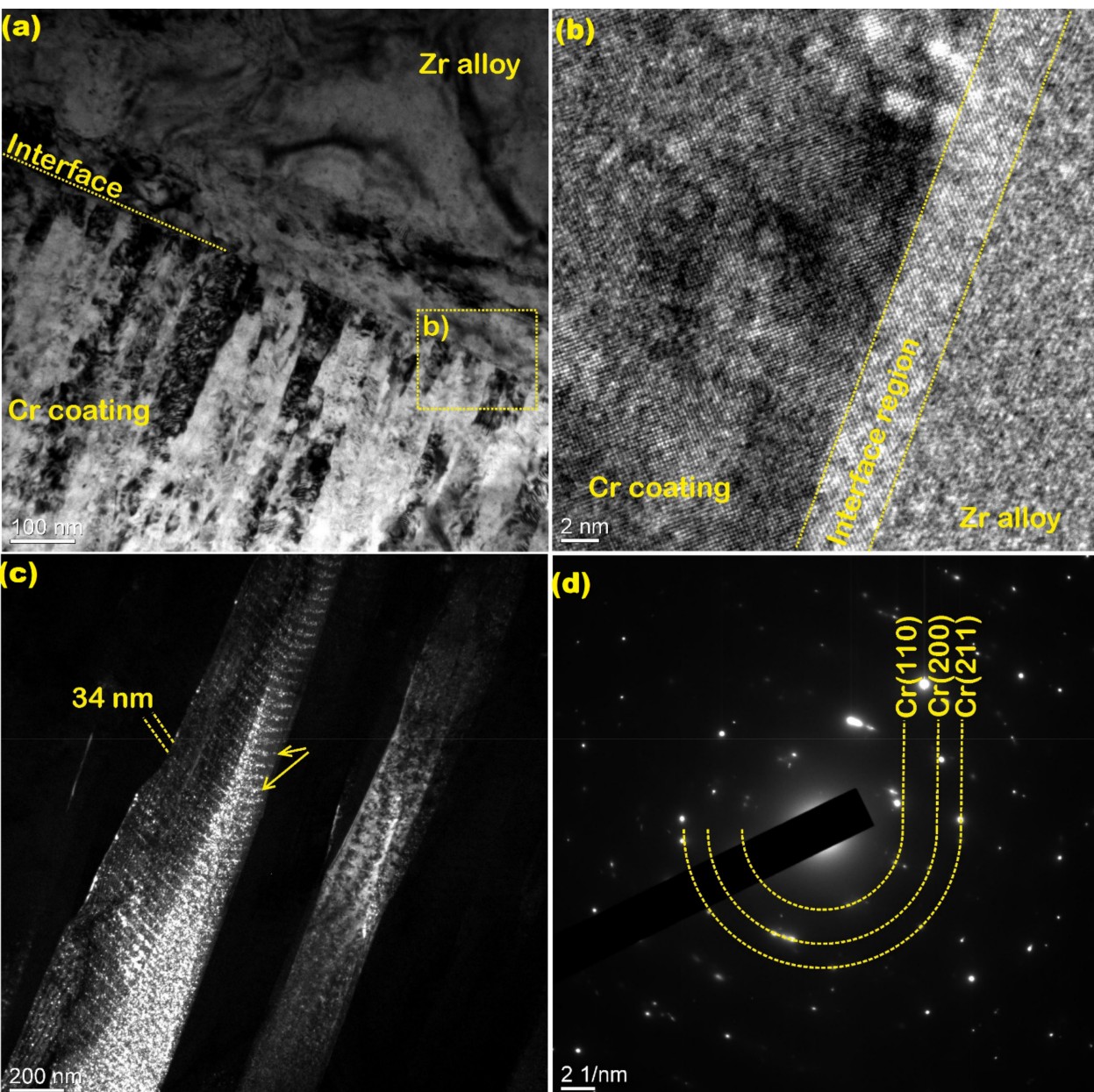

**Figure 5.** TEM images of the Cr coating deposited on Zr alloy substrate (deposition mode 4): (**a**,**b**)—cross-sectional microstructure in interface region; (**c**)—cross-sectional microstructure in Cr coating region; (**d**)—SAED pattern.

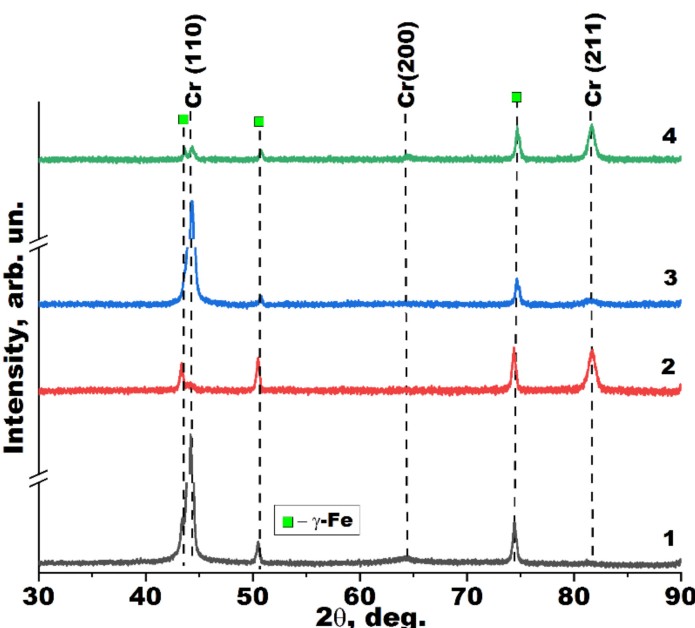

**Figure 6.** XRD diffractograms of Cr coatings obtained on AISI 321 steel substrates depending on their deposition modes.

**Table 2.** Structural parameters of Cr coatings.

| Deposition Mode | $a$, Å | $D$, nm | $T_{C(110)}$ | $T_{C(200)}$ | $T_{C(211)}$ |
|---|---|---|---|---|---|
| 1 | 2.866 | 23 | 2.6 | 0.3 | 0.1 |
| 2 | 2.855 | 10 | 0.2 | 0 | 2.8 |
| 3 | 2.880 | 19 | 2.6 | 0 | 0.4 |
| 4 | 2.879 | 12 | 0.4 | 0.9 | 1.7 |

Note: $a$—lattice parameter; $D$—size of coherent scattering region; $T_{C(110)}$, $T_{C(200)}$ and $T_{C(211)}$—texture coefficients of Cr(110), Cr(200) and Cr(211) planes, respectively.

Table 2 also shows the influence of deposition modes on the size of the coherent scattering region ($D$). It is usually known as the crystallite size according to the Debye Scherrer equation [29]. As RF-ICP assistance was applied, the decrease in $D$ from 23 to 10 and 19 to 12 nm was observed for MS with conventional and hot Cr targets, respectively.

### 3.3. Corrosion Parameters

Figure 7 shows potentiodynamic curves of Cr coatings depending on their deposition modes. The corrosion resistance was evaluated using electrochemical parameters $E_{corr}$, $j_{corr}$ and $R_p$ as determined from graphical extrapolation of polarization curves and using the Stern–Geary equation for calculating $R_p$ [30]. The corrosion rate ($CR$) was determined according to the standard ASTM G 102-89 [31]. Corrosion parameters are summarized in Table 3.

It is clear that the corrosion current density of the uncoated AISI 321 steel ($6.2 \times 10^{-7}$ A/cm$^2$) was higher by one order of magnitude than that of Cr-coated samples. Moreover, a difference in the corrosion current density of Cr coatings could also be observed. It is well known that the coating microstructure can play a key role in corrosion properties [32,33]. According to Table 3, $j_{corr}$ of Cr coatings decreased as RF-ICP assistance was applied for the coating process. For these coatings, the corrosion current density was reduced from $2.8 \times 10^{-8}$ to $1.8 \times 10^{-8}$ and from $2.5 \times 10^{-8}$ to $3.6 \times 10^{-9}$ A/cm$^2$ for MS with conventional and hot Cr targets, respectively. Polarization resistance ($R_p$) of samples was calculated using Tafel slopes ($\beta_\alpha$, $\beta_c$) that displayed the opposite behavior to corrosion current density. Lower values of $R_p$ (2.2 and 5.9 MΩ·cm$^2$) were determined for Cr coatings obtained only by MS, while higher polarization resistance (10.6 and 11.2 MΩ·cm$^2$) was observed for coatings deposited using MS enhanced by RF-ICP source.

Calculations of *CR* revealed the strong effect of Cr coating deposition on corrosion of AISI 321 steel substrates. The uncoated substrate had $6.23 \times 10^{-6}$ mm/year, while Cr-coated steel substrates were characterized by lower values of *CR* in a range of $4.0 \times 10^{-8}$ to $2.8 \times 10^{-7}$ mm/year depending on coating deposition mode.

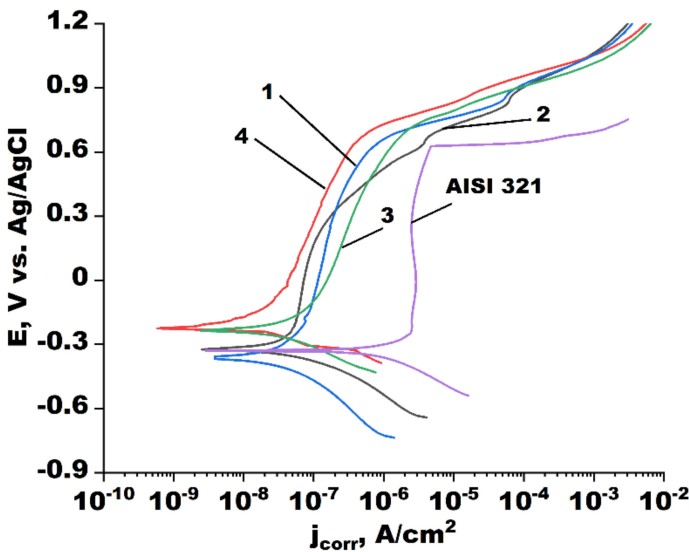

**Figure 7.** Potentiodynamic curves of Cr coatings obtained on AISI 321 steel substrates depending on their deposition modes.

**Table 3.** Corrosion parameters of Cr coatings.

| Deposition Mode | $E_{corr}$, mV | $j_{corr}$·$10^{-9}$ A/cm² | $\beta_a$, mV | $\beta_c$, mV | $R_p$, MΩ·cm² | $CR$, $10^{-6}$ mm/year |
|---|---|---|---|---|---|---|
| AISI 321 | −328 | 621.0 | −155 | 149 | 3.2 | 6.23 |
| 1 | −359 | 27.5 | −140 | 223 | 5.9 | 0.28 |
| 2 | −323 | 17.7 | −141 | 204 | 11.2 | 0.18 |
| 3 | −234 | 24.5 | −89 | 296 | 2.2 | 0.25 |
| 4 | −225 | 3.6 | −53 | 126 | 10.6 | 0.04 |

Note: $E_{corr}$—corrosion potential; $j_{corr}$—corrosion current density; $\beta_a$ and $\beta_c$—anodic and cathodic Tafel slopes, respectively; $R_p$—polarization resistance; $CR$—corrosion rate.

## 4. Discussion

It is well known that structural properties of magnetron-deposited coatings strongly depend on deposition conditions that can be described by particle flux and ion current densities in a substrate [13,34]. Substrate temperature, adatom mobility and other process parameters will be affected by the coating deposition process or using additional plasma or energy sources, e.g., external heater or ion sources. Of course, it will influence coating growth and its properties. For magnetron sputtering, distributions of particle flux and ion current densities of planetary-rotated substrates are non-uniform, as shown in Figures 2 and 3. Maximum values of deposition rate and ion current were observed in the position opposite to the MS. The coating deposition process by can be modified by magnetron sputtering using an RF-ICP source as an additional plasma source for ion assistance. Using calculated and experimental data of particle flux and ion current densities, an ion-to-atom ratio ($N_{i-a}$) of flux entering a substrate was calculated by the following equation:

$$N_{i-a} = \frac{j_{bias}}{t \cdot q \cdot F_{dep}}, \tag{4}$$

where $t$—time. Then, the dependence of $N_{i-a}$ on substrate position was plotted for all deposition modes in Figure 8. It should be mentioned that the obtained data (in Figure 8) should be considered, taking into account parallel positions of substrates to Cr target surface for all substrate positions in planetary-rotated substrate-holders.

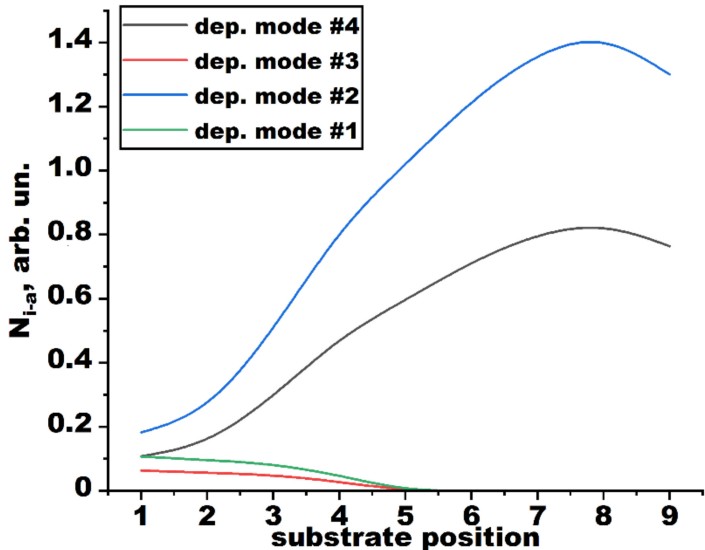

**Figure 8.** Ion-to-atom ratio vs. substrate positions for different deposition modes of Cr coatings.

Different behavior of $N_{i-a}$ was found for Cr coating deposition depending on deposition modes. Indeed, when Cr coatings were obtained only by MS, the ion-to-atom ratio was equal to 0.11 and 0.06 in the position #1 for the MS equipped with conventional and hot Cr targets, respectively. The higher the number of substrate position in the substrate-holder, the less $N_{i-a}$ was observed as the particle flux and ions predominantly entered substrates in positions near the MS. The opposite behavior was found in the case of MS enhanced by RF-ICP assistance. The ion-to-atom ratio significantly increased to 1.43 and 0.84 as the substrate was moved farther away from the MS with conventional and hot Cr targets (pos. #1→#8–9), respectively. Moreover, the increase in $N_{i-a}$ (0.18 and 0.11) was even observed in position #1, when RF-ICP assistance was applied. This change in the ion-to-atom ratio results in modifying coating properties. A typical columnar microstructure of Cr coatings was observed for magnetron sputtering in modes #1 and #3 as deposition conditions of coatings corresponded to zone 1 in SZM models [14,15]. While Cr coatings with a dense cross-sectional microstructure can be obtained under ion assistance formed by the RF-ICP source (Figure 4). Indeed, densification of coating microstructure can be performed using intensive ion bombardment during magnetron sputtering that is usually achieved by applying MS with unbalanced magnetic field configuration [35,36] or high discharge currents in a pulsed mode (HiPIMS, e.g., in [1,17,37]).

More detailed structural analysis of the Cr coating using RF-ICP assistance (dep. mode #4) revealed the formation of a layered structure (Figure 5). This structure was formed due to changes in ion-to-atom ratio in the case of planetary rotation of a substrate. Based on $N_{i-a}$ calculations in Figure 8, two parts of the coating process can be highlighted in the case of magnetron deposition enhanced by the RF-ICP source. The first is a deposition near the MS, when the ion-to-atom ratio was relatively low (~0.1–0.4 in positions #1→#4). The second is characterized by a low value of $F_{dep}$, which is accompanied with some ion bombardment initiated by substrate biasing in a RF-ICP discharge. Due to this, the flux entered a substrate in positions #5→#9 and has high $N_{i-a}$ values. All these data highlight the two-step process of coating deposition including coating deposition and further coating bombardment for a single whole rotation. Indeed, the observed thickness (~34 nm) shown in Figure 5 corresponds well with the thickness of the layer on a planetary-rotated substrate during coating deposition for a whole turn.

Despite densification of the coating microstructure, the crystal structure of Cr coatings can be changed with ion assistance. According to XRD data (Table 2), changes in texture coefficients were observed from Cr (110) to Cr (211), when RF-ICP assistance was applied. The Cr (110) texture forms more easily [38] in the case of a high number of nucleated islands on the substrate surface as the lowest surface energy belongs to the Cr (110) direction in bcc Cr [38,39]. Using RF-ICP assistance leads to a change in conditions of grain growth along the Cr (211) direction for magnetron sputtering ($T_{C(211)}$~1.7). In this case, adatom mobility on the substrate surface and the "knock-on" effect increased [28], potentially resulting in grain growth kinetics. Indeed, there was less texturing of Cr coatings along the Cr (211) plane ($T_{C(211)}$~2.8) for the MS enhanced by the RF-ICP source, since a higher deposition rate and lower ion-to-atom ratio were two of these deposition conditions.

Cr coating deposition can improve corrosion behavior of AISI 321 steel substrates, since lower *CR* values were observed for coated substrates. It was shown that corrosion rate can be reduced by one order of magnitude ($6.2 \times 10^{-6} \to 2.8 \times 10^{-7}$ mm/year) when Cr coatings were prepared by magnetron sputtering. The increase in corrosion resistance of Cr-coated steel samples has also been shown in published papers [1,32]. As a result of the RF-ICP assistance, the corrosion resistance of Cr coatings having denser coating microstructures became higher than that of coatings obtained using only magnetron sputtering. This is in good agreement with the published data [20,40–42]. Moreover, texturing of Cr coatings along the Cr (211) direction with RF-ICP assistance can improve corrosion parameters as coatings with Cr (110) grain directions are usually characterized by higher corrosion current and/or oxidation rate [43,44].

Thus, this study highlights the significant enhancement of cross-sectional microstructure and corrosion parameters of Cr coatings obtained using MS enhanced by RF-ICP assistance.

## 5. Conclusions

This comparative study of Cr coating deposition using a magnetron sputtering enhanced by a RF-ICP source revealed the following:

1. RF-ICP assistance can increase the ion current density of a substrate by almost twofold when magnetron sputtering is applied in the coating process. It causes an increase in ion-to-atom ratio in the flux of a substrate.
2. Calculations of particle and ion flux densities showed a change in the dependence of ion-to-atom ratio on substrate position, when planetary rotation of substrates was used. Magnetron deposition enhanced by an RF-ICP source became a two-step process including stages with relatively low (0.11–0.18) and high (0.84–1.43) ion-to-atom ratios in particle flux entering a substrate.
3. Intensive ion bombardment during the coating process induced by RF-ICP assistance can result in modification of the crystal structure of Cr coatings from Cr (110) to Cr (211) orientations and coating densification. Due to the two-stage deposition process, Cr coating can have a layered structure caused by ion bombardment in positions distant from the magnetron sputtering system.
4. Cr coating deposition led to a decrease in the corrosion rate of AISI 321 steel from $6.2 \times 10^{-6}$ to $2.8 \times 10^{-7}$ mm/year in a 3.5 wt.% NaCl solution. The corrosion rate of Cr-coated steel substrates can be reduced up to $4.0 \times 10^{-8}$ mm/year by applying RF-ICP assistance for magneton sputtering.

**Author Contributions:** Conceptualization, D.V.S.; Methodology, D.V.S.; Validation, D.V.S., G.A.B.; Formal analysis, D.V.S., V.A.G. and G.A.B.; Investigation, D.V.S., V.A.G., K.A.Z., K.A.; Resources, D.V.S. and G.A.B.; Data curation, V.A.G.; Writing—original draft preparation, D.V.S. and V.A.G.; Writing—review and editing, D.V.S.; Visualization, D.V.S. and V.A.G.; Supervision, D.V.S.; Project administration, D.V.S.; Funding acquisition, D.V.S. All authors have read and agreed to the published version of the manuscript.

**Funding:** This study was supported by a grant from the President of the Russian Federation to support young Russian scientists, project number MK-3570.2022.4.

**Institutional Review Board Statement:** Not applicable.

**Informed Consent Statement:** Not applicable.

**Data Availability Statement:** Not applicable.

**Acknowledgments:** The authors thank the CSU NMNT TPU for use of its TEM equipment (JEF-2100) supported by RF MES project #075-15-2021-710.

**Conflicts of Interest:** The authors declare no conflict of interest.

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
