# Peer review of "Magnetron Deposition of Cr Coatings with RF-ICP Assistance"

_coatings, doi:10.3390/coatings12101587_

Round 1
Reviewer 1 Report
The manuscript entitled "Magnetron deposition of Cr coatings with RF-ICP assistance" is interesting and well written. The subject addressed in this article is worthy of investigation. The conclusions were supported by the data. The focus of the manuscript is within scope of the journal but the authors have to make some minor corrections and amendments before paper is acceptable for publication in journal Coatings. Following are my specific remarks:
1. Page 1, line 45. After "radiofrequency inductively coupled plasma" an acronym RF-ICP should be put in parenthesis.
2. Page 4. Please specify the working area of the electrode, temperature, and scan rate for the potentiodynamic polarization method.
3. Page 5, line 172. Erase the green line.
4. Page 8, lines 241-243. Rephrase the sentence. The corrosion resistance was evaluated using the electrochemical parameters Ecorr, jcorr and Rp as determined from graphical extrapolation of polarization curve and using the Stern-Geary equation for calculating Rp and the results are summarized in Table 3.
5. Page 9, Fig. 7. Correct the Vertical Axis Title to read "E, V vs. Ag/AgCl" rather than "E, V".
6. Page 9, Fig. 7. Each individual line must be more precisely rendered.
7. Page 9, Table 3. Please check Ecorr of Cr coating on AISI 321 steel substrates prepared by magnetron sputtering of conventional Cr targets without RF-ICP assistance (deposition mode 1).
Author Response
Dear Reviewer,
many thanks for your remarks. We improved our manuscript. All changes are highlighted by green in the re-submitted article.
Note #1: Page 1, line 45. After "radiofrequency inductively coupled plasma" an acronym RF-ICP should be put in parenthesis.
Response #1: Thank you for the remark. We corrected the sentence (the line 49).
Note #2: Page 4. Please specify the working area of the electrode, temperature, and scan rate for the potentiodynamic polarization method.
Response #2: Thank you for the remark. We added the data in the part 2.3 (the lines 146-148 and 150-152).
Note #3: Page 5, line 172. Erase the green line.
Response #3: Thank you for the remark. We deleted this mistake.
Note #4: Page 8, lines 241-243. Rephrase the sentence. The corrosion resistance was evaluated using the electrochemical parameters Ecorr, jcorr and Rp as determined from graphical extrapolation of polarization curve and using the Stern-Geary equation for calculating Rp and the results are summarized in Table 3.
Response #4: Thank you for the remark. We rephrased the sentence (the lines 251-255).
Note #5: Page 9, Fig. 7. Correct the Vertical Axis Title to read "E, V vs. Ag/AgCl" rather than "E, V".
Response #5: Thank you for the remark. We corrected Y-axis.
Note #6: Page 9, Fig. 7. Each individual line must be more precisely rendered.
Response #6: Thank you for the remark. We modified the Fig. 7.
Note #7: Page 9, Table 3. Please check Ecorr of Cr coating on AISI 321 steel substrates prepared by magnetron sputtering of conventional Cr targets without RF-ICP assistance (deposition mode 1).
Response #7: Thank you for the remark Indeed, Ecorr of this sample was corrected.
Best wishes,
Dmitrii V. Sidelev
PhD., Associate Prof.
Tomsk Polytechnic University
phone: +7-3822-70-17-77 (add. 2518)
mob. phone: +7-983-238-71-79

Reviewer 2 Report
Dear Authors,
I read with pleasure your manuscript, which remind me the dawn of my career.
It is well done, well written, and of outstanding scientific level. Everything is thorougly described, and of high pedagogic level.
I want to accept as it: I have only minor comments that I am going to listen below:
Line 51-52: I would avoid to speak about CrNx and CuOx. and removing the references (cheap self-citations)
Line 128-131: the references to project can be added in the aknowledgements.
I have a question: can you estimate the temperature of the uncooled Cr target?
Figure 4: says what 1, 2, 3, and subfigure 4 are.
My best regards.
Author Response
Dear Reviewer,
many thanks for your remarks. We improved our manuscript. All changes are highlighted by green in the re-submitted article.
Note #1: Line 51-52: I would avoid to speak about CrNx and CuOx and removing the references (cheap self-citations)
Response #1: Thank you for the remark. We deleted these refs.
Note #2: Line 128-131: the references to project can be added in the aknowledgements.
Response #2: Thank you for the remark. We transferred the reference on acknowledgment from part 2.3 to the part “Acknowledgements”.
Note #3: I have a question: can you estimate the temperature of the uncooled Cr target?
Response #3: Thank you for the remark. We calculated the temperature of the hot Cr target in the previous papers [10.1016/j.surfcoat.2016.06.096; 10.1016/j.vacuum.2016.07.030].
Note #4: Figure 4: says what 1, 2, 3, and subfigure 4 are.
Response #4: Thank you for the remark. We corrected the title of the Fig. 4 as “Figure 4. SEM images of cross-section microstructure of Cr coatings obtained on Si (110) substrates depending on their deposition modes: 1 – dep. mode #1; 2 – dep. mode #2; 3 – dep. mode #3; 4 – dep. mode #4.”.
Best wishes,
Dmitrii V. Sidelev
PhD., Associate Prof.
Tomsk Polytechnic University
phone: +7-3822-70-17-77 (add. 2518)
mob. phone: +7-983-238-71-79

Reviewer 3 Report
The manuscript presents an interesting study about the deposition of Cr coatings with radiofrequency inductively coupled plasma assistance on Si wafers and steel. However, the paper needs major revisions before it is processed further, some comments follow:
Abstract:
The abstract must be improved. The abstract must contain information about:
- Background: Please highlight the novelty of the study;
- Methods: Describe briefly the main methods used to obtain and characterize the coating.
- Results and conclusions: Indicate the main conclusions or interpretations, also add some quantitative results.
Introduction section
The introduction section must be improved. In the last paragraph of the introduction section please highlight the novelty and the aim of this study, also add information about the characterization methods used and the main conclusion.
Materials and methods
Line 71. Please introduce a table with the chemical composition of the steel.
Lines 135-139. Please introduce the information regarding the surface exposed, potential range etc. Also, introduce the software used to acquire and process the experimental data.
Results
Figure 4. Please introduce figure labels to highlight the interest zone for the reader.
Table 3. Please calculate the corrosion rate and introduce the values in the table. Also, discuss them. See an example: DOI: 10.3390/ma13153410
Discussion
The discussion section must be improved. Compare the results obtained with other coatings and discuss them. Also, regarding the corrosion behaviour, it is important to compare the results with the corrosion behaviour of the steel used. Therefore, please introduce the values of the polarization test made on the steel used as a substrate or, if it is studied already, introduce some values and citations and reveal the improvement of the corrosion resistance by coating.
Conclusions
The conclusion looks vague. The conclusion can be written by points. Add quantitative results and also limitations and suggestions.
References
There are too many self-citations (over six). Please remove them.
Author Response
Dear Reviewer,
many thanks for your remarks. We improved our manuscript. All changes are highlighted by green in the re-submitted article.
Note #1: The abstract must be improved. The abstract must contain information about:
- Background: Please highlight the novelty of the study;
- Methods: Describe briefly the main methods used to obtain and characterize the coating.
- Results and conclusions: Indicate the main conclusions or interpretations, also add some quantitative results.
Response #1: Thank you for the remark. We modified the abstract.
Note #2: The introduction section must be improved. In the last paragraph of the introduction section please highlight the novelty and the aim of this study, also add information about the characterization methods used and the main conclusion.
Response #2: Thank you for the remark. We corrected the part “Introduction” (the lines 55-65).
Note #3: Line 71. Please introduce a table with the chemical composition of the steel.
Response #3: Thank you for the remark. We added composition of the steel (the lines 80-82).
Note #4: Lines 135-139. Please introduce the information regarding the surface exposed, potential range etc. Also, introduce the software used to acquire and process the experimental data.
Response #4: Thank you for the remark. We added the information about corrosion tests (the lines 146-148 and 150-152).
Note #5: Figure 4. Please introduce figure labels to highlight the interest zone for the reader.
Response #5: Thank you for the remark. We modified the title of the Fig. 4 as “Figure 4. SEM images of cross-section microstructure of Cr coatings obtained on Si (110) substrates depending on their deposition modes: 1 – dep. mode #1; 2 – dep. mode #2; 3 – dep. mode #3; 4 – dep. mode #4.”.
Note #6: Table 3. Please calculate the corrosion rate and introduce the values in the table. Also, discuss them. See an example: DOI: 10.3390/ma13153410.
Response #6: Thank you for the remark. We calculated CR and added in the Table 3. Then, we discussed this new data (the lines 273-276).
Note #7: The discussion section must be improved. Compare the results obtained with other coatings and discuss them. Also, regarding the corrosion behaviour, it is important to compare the results with the corrosion behaviour of the steel used. Therefore, please introduce the values of the polarization test made on the steel used as a substrate or, if it is studied already, introduce some values and citations and reveal the improvement of the corrosion resistance by coating.
Response #7: Thank you for the remark. We added the potentiodynamic curve for the uncoated AISI 321 steel in Fig. 7 and, then, calculated corrosion parameters and added it in the Table. 3.
After it, we added some discussion about the role of Cr coating on corrosion properties (the lines 312-315 and 340-344).
Note #8: The conclusion looks vague. The conclusion can be written by points. Add quantitative results and also limitations and suggestions.
Response #8: Thank you for the remark. We corrected the conclusion.
Note #9: There are too many self-citations (over six). Please remove them.
Response #9: Thank you for the remark. We deleted the ref. 24 and 25.
Best wishes,
Dmitrii V. Sidelev
PhD., Associate Prof.
Tomsk Polytechnic University
phone: +7-3822-70-17-77 (add. 2518)
mob. phone: +7-983-238-71-79

Round 2
Reviewer 3 Report
The manuscript can be published in the present form